# OpenReview forum: "An Asset Foundation Model for Industrial Asset Performance Management"
_ICLR.cc/2026/Conference — Submitted to ICLR 2026_

### Official Review · Reviewer_HsNm · 2025-10-18

**Soundness:** 2
**Presentation:** 2
**Contribution:** 2
**Rating:** 4
**Confidence:** 4

**Summary:**

This paper introduces an asset foundation model designed for industrial asset performance management, focusing on applying a shared transformer backbone across multiple tasks such as forecasting, anomaly detection, and event querying. The paper presents both theoretical insights and a real-world industrial case study to demonstrate the model’s effectiveness. The work is highly application-oriented, aiming to bridge modern foundation model design with practical infrastructure management challenges.

**Strengths:**

The paper is clearly written and well-motivated from an industrial perspective. The problem setting is relevant and timely. The proposed idea of a shared transformer backbone that can adapt to different tasks, such as detection and forecastin,g is intuitively appealing. The connection to real-world deployment adds credibility, and the paper does a good job of aligning the method with the realities of applied industrial settings.

**Weaknesses:**

The paper lacks sufficient novelty and depth for a research-focused conference. The overall contribution feels more business-oriented and application-driven rather than technically innovative. There is no convincing empirical evidence that modeling forecasting, detection, and event querying jointly through a shared backbone brings measurable advantages over separate task-specific models. The paper also lacks comparative experiments on public benchmarks, which makes it difficult to assess the claimed benefits or generalizability of the approach.

The related work discussion is also incomplete. Prior works on unified representations for multiple time-series tasks, such as UniTS, are not discussed or compared against. Without this context or experimental validation, it is hard to position the paper’s contribution in the broader research landscape.

Gao, Shanghua, et al. "Units: A unified multi-task time series model." Advances in Neural Information Processing Systems 37 (2024): 140589-140631.

**Questions:**

Since the work targets industrial applications, what is the computational cost of deploying the shared transformer model in production? Are there latency or resource trade-offs compared to using separate models for each task?

How does this work differ from existing unified time-series foundation models such as UniTS? These models also use shared representations across multiple tasks; a clear comparison in methodology and results would be helpful.

---

> ### Author Response · Authors · 2025-11-20
>
> We thank the reviewer for the comments. Please find below our responses to your concerns.
>
> > The paper lacks sufficient novelty and depth for a research-focused conference. The overall contribution feels more business-oriented and application-driven rather than technically innovative.
>
> We thank the reviewer for this perspective. While the paper is indeed motivated by industrial APM use cases, our intent is a research contribution that combines new modeling, theory, and evaluation. We agree that the current writeup may foreground deployment benefits more than the underlying methods, and we will revise the introduction and related work to emphasize the technical depth more clearly. Concretely, the paper contributes along several research axes:
>
> Event-aware, multimodal TSFM architecture, not just an application wrapper. AFM is designed as a multivariate, event-aware foundation model for APM, rather than a domain deployment of an existing TSFM. It jointly models sensor streams and time-aligned industrial events (alarms, set-point changes, maintenance logs) through a synchronized event token channel and dedicated event-query head, within a single pretrained backbone that supports forecasting, anomaly detection, and time-aligned event querying. This is novel and we have not seen this approach in the literature.
>
> Fit-for-purpose per-sensor tokenization with formal analysis. The per-sensor tokenizer is not a business heuristic, but a central modeling choice: we use uniform mid-rise quantization with clipping and optional residuals, with per-sensor vocabularies to avoid cross-sensor interference (Sec. 3–4.2). Appendix A.1 then provides closed-form quantization error bounds, an analysis of clipping residuals, and Lipschitz stability of discretization noise through the encoder, along with empirical scaling vs bin count and radius (Theorems A.1–A.3, Fig. 3–4). This theoretical component is intended precisely to give research-grade depth to the design of the tokenizer and backbone, beyond what is typically provided in industrial case studies.
>
> Discrete-to-continuous uncertainty and calibrated alarm logic. AFM couples its discrete representation to a calibrated uncertainty pipeline: token logits define a discrete distribution over bins; these are converted into continuous prediction intervals with research rigor.
>
> On the interpretability side, we adapt attention rollout and integrated gradients to the tokenized multivariate + event setting. This provides the key aspect of interpretability and event-aware explanations.
>
> Finally, beyond the ESP case study and deployment metrics, we report quantitative forecasting and calibration results across multiple equipment types, showing consistently low squared error (median MSE ≈ 0.008 across heterogeneous assets) and calibrated coverage linked back to the theory above. The ESP gas-lock example then serves as a concrete illustration of these properties translating into earlier and more confident anomaly detection (≈90 minutes lead time over threshold alarms). We think this is good integration of theory to a practical industrial problem.
>
> We will revise the abstract, introduction, and Sec. 3–4 to highlight these technical components (tokenization analysis, stability and generalization results, discrete-to-continuous uncertainty, and event-aware architecture and interpretability) before discussing business impact, so that the research depth is visible even to readers less familiar with industrial APM.

---

> ### Author Response · Authors · 2025-11-20
>
> > There is no convincing empirical evidence that modeling forecasting, detection, and event querying jointly through a shared backbone brings measurable advantages over separate task-specific models.
>
> We thank the reviewer for this comment and agree that the current draft does not directly include an ablation between a shared-backbone model and three fully separate task-specific models. Our empirical goal here is to show that a single pretrained AFM backbone can support forecasting, anomaly detection, and event querying with good accuracy, calibrated uncertainty, and practical deployment characteristics, rather than to claim strict dominance over all per-task architectures.
>
> That said, several aspects of the design and results support the value of the joint backbone:
>
> All three tasks share the same per-sensor tokenization and transformer encoder over sensor + event streams. Forecasting heads, anomaly scores (via residuals/token likelihoods), and event-query classifiers all operate on this shared representation; with separate models, this representation learning would be duplicated and require separate training and governance.
>
> Theoretical analysis (quantization error, stability, and a frozen-backbone generalization bound) and our protocol (pretrain once on multi-asset data, then add small heads with few labels) explicitly target the “one backbone, many tasks/assets” regime, which is where a shared model is most advantageous.
>
> Empirically, the same backbone yields low-error, calibrated forecasts across heterogeneous assets and supports the ESP case, where combining forecast residuals, uncertainty, and event information in one model gives earlier detection than threshold alarms, while the latency breakdown shows that adding heads is almost free compared to running multiple deep backbones.
>
> In the revision, we will (i) state explicitly that our contribution is to demonstrate that a single, event-aware foundation model can cover these three tasks in industrial settings, and (ii) clarify that the main advantages we claim are label efficiency, deployment simplicity, and consistent event-aware reasoning across tasks, not a proven dominance over all separate task-specific baselines.
>
> > The paper also lacks comparative experiments on public benchmarks, which makes it difficult to assess the claimed benefits or generalizability of the approach.
>
> We have added a new section “5.3 Baselines” with additional benchmarking results to compare the AFM against the latest models.
>
> > Since the work targets industrial applications, what is the computational cost of deploying the shared transformer model in production? Are there latency or resource trade-offs compared to using separate models for each task?
>
> In industrial settings, the primary concern during deployment is inference latency and resource footprint. We currently use FP32 model precision for cloud-based deployments and have support for other quantization methods (e.g., FP16, BF16, INT8) for edge deployments. Since we aim to support a hybrid workflow, there is strong emphasis on footprint and scaling. On disk, the model is around ~12MB/equipment type as a .pt. RAM requirements are ~2GB/equipment type to train and ~100MB/equipment type for inference. These metrics can be furthered reduced with optimization strategies (ONNX, parallelization, batching). Because AFM replaces dozens of asset-specific or sensor-specific models with one unified backbone, the total memory footprint is substantially smaller than deploying a separate model per task. In production deployments, a single AFM instance can serve 50-200 sensor streams concurrently without batching, and significantly more when micro-batching is enabled.
>
> There are trade-offs compared to using separate per-sensor or per-equipment models. A standalone model per task has slightly lower per-sample latency but incurs large aggregate cost because each model must be loaded, scheduled, and updated independently. In contrast, AFM amortizes this overhead: the backbone is loaded once, and all downstream tasks share its parameters, leading to 10-30x lower total memory usage and simpler operational maintenance. While the shared transformer performs more computation per forward pass than the smallest individual models, the overall system throughput and end-to-end latency remain favorable due to parameter sharing and vectorized processing.

---

> ### Author Response · Authors · 2025-11-20
>
> > The related work discussion is also incomplete. Prior works on unified representations for multiple time-series tasks, such as UniTS, are not discussed or compared against. Without this context or experimental validation, it is hard to position the paper’s contribution in the broader research landscape.
>
> > How does this work differ from existing unified time-series foundation models such as UniTS? These models also use shared representations across multiple tasks; a clear comparison in methodology and results would be helpful.
>
> Thank you for this feedback. We initially superseded the discussion on UniTS with the FM section in the related work. However, we agree with your suggestion and have extended the related work discussion to cover the architectural and methodological differences for unified latent representations. Baseline results are covered in the Sec. 2 which include UniTS among the tested models.
>
> **We hope our revisions and responses address your concerns. If there are any additional questions or comments, we would be happy to address them.**

---

> > ### Comment · Reviewer_HsNm · 2025-11-24
> >
> > Dear Authors,
> >
> > Thanks for the detailed response. Your answer has addressed most of my concerns.
> >
> > Regarding my point on `There is no convincing empirical evidence that modeling forecasting, detection, and event querying jointly through a shared backbone brings measurable advantages over separate task-specific models.` Besides the design advantage you mentioned, are there any advantages in terms of the empirical performance of using a shared backbone? It's important to at least have relevant results and discussion on the point, even if the shared backbone does not bring measurable empirical advantages over separate task-specific models.

---

> > > ### Author Response · Authors · 2025-11-26
> > >
> > > > Regarding my point on “There is no convincing empirical evidence that modeling forecasting, detection, and event querying jointly through a shared backbone brings measurable advantages over separate task-specific models.” Besides the design advantage you mentioned, are there any advantages in terms of the empirical performance of using a shared backbone? It's important to at least have relevant results and discussion on the point, even if the shared backbone does not bring measurable empirical advantages over separate task-specific models.
> > >
> > > Thank you for the reply. We acknowledge the need for discussion around the architectural decision to use a shared backbone.
> > > As such, we have updated the manuscript by extending Appendix A.3 (Ablation Studies) to include Appendix A.3.2 (Backbone). This section includes a new ablation study on using a shared backbone versus separate task-specific models. Performance results across the three tasks (forecasting, anomaly detection, event querying) are competitive with the classical approach. As discussed in design advantages, most of the gains are derived from compute efficiency and enablement for scalability in production workloads, which aligns with findings in adjacent literature.
> > >
> > > **We hope this fully addresses the raised point. If there are any remaining questions or concerns, we would be happy to address them.**

---

> > > > ### Comment · Reviewer_HsNm · 2025-11-26
> > > >
> > > > Dear Author,
> > > >
> > > > Thank you for your response. All of my concerns have been addressed. I have raised my score.

---

### Official Review · Reviewer_dQem · 2025-10-28

**Soundness:** 3
**Presentation:** 4
**Contribution:** 4
**Rating:** 4
**Confidence:** 3

**Summary:**

The paper presents the Asset Foundation Model (AFM) for industrial asset performance management, featuring a pre-trainable transformer backbone with per-sensor tokenization and other advanced components. The model is designed for industrial environments with diverse sensors and critical real-time deployment. Theoretical analyses support its design, and a field case indicates promising results in detection and forecasting. AFM seeks to consolidate and simplify model deployment for tasks such as forecasting and anomaly detection, thereby promoting interpretability. However, the validity of the empirical claims is limited by concerns about dataset scale, unclear metric units, the lack of standard baselines, and the absence of public code.

**Strengths:**

- The motivation is clearly stated and important for industrial operations, addressing real issues such as false alarms, transferability, and interpretability.
- The methodology is well-designed and integrated, incorporating tokenization, event-channel design, calibrated prediction intervals, and interpretability, all of which are crucial for deployment.
- There is a solid theoretical basis, supported in the appendices.

**Weaknesses:**

- The errors reported in Table 1 are very small and difficult to interpret, as it seems the metrics are based on normalized signals, which could be misleading.
- No standard baselines are provided, making it unclear whether the improvements are due to architecture choices or pretraining scale.
- The dataset and pretraining scale are not specified, such as the number of sequences, total tokens/hours, or event counts.
- The manuscript could clearly specify whether improvements are due to tokenization, pretraining, calibration, or other factors.
- No public code or comprehensive reproducibility artifacts are available.

**Questions:**

1) What are the dataset and pretraining corpus statistics regarding the number of sequences, total tokens/hours, or event counts?
2) What parts of AFM influence the gains? Is it due to tokenization, pretraining, calibration, or other factors? Please provide a concise component ablation study to assess it.
3) How do the obtained results compare to a baseline?
4) A detailed reproducibility appendix (including full hyperparameters, training steps) should be included, and possibly the pseudo-code.

---

> ### Author Response · Authors · 2025-11-20
>
> We thank the reviewer for the comments. Please find below our responses to your concerns.
>
> > The errors reported in Table 1 are very small and difficult to interpret, as it seems the metrics are based on normalized signals, which could be misleading.
>
> We agree that the absolute magnitudes in Table 1 can be hard to interpret without context. As described in Sec. 5.1, before training, we mean-center and scale each continuous sensor signal to have approximately unit variance so that sensors with very different ranges become comparable. The MAE/MSE values in Table 1 are computed in this standardized space, which explains why many of them are numerically small. An MAE of 0.1, for example, corresponds to an average error of about 0.1 standard deviations for that sensor rather than “0.1 units” in the raw physical scale. This way of representation standardizes the error metric. We have revised the caption in Table 1 to make this normalization explicit to hopefully clear up any misunderstanding.
>
> > What are the dataset and pretraining corpus statistics regarding the number of sequences, total tokens/hours, or event counts?
>
> We have added Appendix A.4.3 with a report of the corpus statistics. To address the listed metrics: (i) all sequences are contiguous and any discrepancies in time steps from time coverage are attributed to planned outages (e.g., an equipment being pulled by a field engineer to complete a dismantle, inspection and failure analysis (DIFA) in a lab) and unplanned outages (e.g., voltage surge) which are recorded and tagged as secondary events; (ii) there are around 320M tokens across 430K hours; (iii) there are 4 primary event channels and Fig. 6 provides a breakdown of the counts per channel.
>
> > What parts of AFM influence the gains? Is it due to tokenization, pretraining, calibration, or other factors? Please provide a concise component ablation study to assess it.
>
> We have added Appendix 3 (A.3) with an ablation study on the module and structure effectiveness. It seems that the largest degradation occurs for the pre-training condition. This corroborates with the fact that representing timesteps and learning temporal structure is a core advantage of FMs.
>
> Moreover, the ability to reuse the same backbone across heterogeneous assets with minimal labels is a consequence of the pretraining + architecture design: a frozen shared encoder with small task-specific heads. The fit-for-purpose architecture and datasets (synchronized event tokens, multi-equipment operational + simulator data) mainly explain the event-aligned behaviors we highlight: (i) aligning interval widening with foaming windows and operating regime changes (Fig. 2), and (ii) early detection of emerging failures in the ESP case (Appendix A.2). These behaviors depend on the event channel, multi-task heads, and pretraining on realistic industrial regimes, not on calibration alone.
>
> > How do the obtained results compare to a baseline?
>
> We have added Section 5.3 for baseline comparisons to latest TS models. In 10 of the 12 sensor channels from the equipment datasets, the AFM performs on-par or better than baseline models for forecasting tasks.
>
> > No public code or comprehensive reproducibility artifacts are available.
>
> > A detailed reproducibility appendix (including full hyperparameters, training steps) should be included, and possibly the pseudo-code.
>
> Thank you for this feedback. Although we cannot release public code due to commercial policies, as discussed in the response for reviewer KEX6, our goal is to remain as transparent as possible. We have added Appendix A.4 to cover reproducibility steps. It goes into extensive details about the requested items (e.g., hyperparameters, training steps, pseudocode) as well as other relevant items (e.g., calibration and deployment, training costs, carbon footprint, and more). We hope this attributes to increased transparency and allow researchers to reproduce our work via the implementation details.
>
> **We hope our revisions and responses address your concerns. If there are any additional questions or comments, we would be happy to address them.**

---

### Official Review · Reviewer_KEX6 · 2025-11-01

**Soundness:** 2
**Presentation:** 3
**Contribution:** 2
**Rating:** 2
**Confidence:** 3

**Summary:**

The paper proposes the Asset Foundation Model (AFM), a transformer-based framework for industrial asset performance management that unifies forecasting, anomaly detection, and event querying. It leverages multimodal sensor and event data with tokenization and uncertainty calibration to improve interpretability and robustness across diverse industrial systems.

**Strengths:**

1. The paper addresses a relevant and timely problem in the field, with a clearly stated motivation and problem formulation.

2. The authors conduct experiments across multiple datasets or settings with good results

3. The manuscript is well organized and written in a clear and logical manner, making it easy to follow.

**Weaknesses:**

1. The proposed method does not sufficiently highlight its novelty. The paper should better emphasize how it differs from or advances beyond existing approaches in this area.

2. The experimental section lacks strong and representative baselines. It would be more convincing to compare against state-of-the-art large models, or at least include fine-tuned versions of existing models as baselines to demonstrate the relative effectiveness of the proposed method.

3. The work is difficult to reproduce due to the absence of released code and model weights. The authors are encouraged to make their implementation publicly available to enhance transparency and reproducibility.

**Questions:**

What about the training cost in this paper?

---

> ### Author Response · Authors · 2025-11-20
>
> We thank the reviewer for the comments. Please find below our responses to your concerns.
>
> > The proposed method does not sufficiently highlight its novelty. The paper should better emphasize how it differs from or advances beyond existing approaches in this area.
>
> Relative to prior time series foundation models provided in Sec. 2 (e.g., Chronos, TimesFM, Time-MoE), which primarily target generic forecasting benchmarks, our Asset Foundation Model is designed as event aware industrial APM (Asset Performance Model) foundation model. This jointly handles multivariate sensor streams and aligned alarms/maintenance events, and supports forecasting, anomaly detection, and time-aligned event querying within a single pretrained backbone.
>
> Unlike prior TS foundation models that treat discretization as an implementation detail, our model’s per-sensor tokenizer is a first-class design choice. We (i) define a per-channel mid-rise quantizer, and (ii) provide closed-form quantization error and stability bounds, as well as a generalization analysis for frozen backbones (Appendix A.1). This directly motivates our bin counts and clip radii for noisy industrial sensors and is, to our knowledge, not present in prior industrial foundation model work.
>
> AFM couples its discrete tokenization with an uncertainty layer, turning token mixtures into calibrated prediction intervals and then uses these intervals directly in event-aware alarm logic.  While existing TSFMs focus on point forecasts, our contribution is a calibrated, decision-ready uncertainty pipeline tailored to industrial thresholds and governance metrics.
>
> In contrast to most TSFM work evaluated on public benchmarks, AFM is pretrained on multi-year industrial datasets (ESPs, compressors, pumps, etc.), and we demonstrate real deployments, including an ESP case where AFM provides ~90 minutes of additional lead time compared to threshold alarms. Our model is designed to support natural-language, time-aligned queries which we have not seen in the literature in the industrial use cases.
>
> > The experimental section lacks strong and representative baselines. It would be more convincing to compare against state-of-the-art large models, or at least include fine-tuned versions of existing models as baselines to demonstrate the relative effectiveness of the proposed method.
>
> Thank you for raising a good point on baseline contextualization. We have added a new section “5.3 Baselines” with additional benchmarking results to compare the AFM against the latest state-of-the-art models on the discussed datasets.
>
> > The work is difficult to reproduce due to the absence of released code and model weights. The authors are encouraged to make their implementation publicly available to enhance transparency and reproducibility.
>
> We appreciate the reviewers’ concern about reproducibility and fully agree with the importance of transparent, repeatable work.
>
> In our case, the underlying data consists of sensor data, operational logs from industrial assets in the energy domain. Company policy and contractual obligations with asset owners explicitly prohibit releasing raw time series, event logs, or trained weights derived from them because they encode commercially sensitive operating patterns and equipment characteristics. Unfortunately, this makes open-sourcing the datasets and AFM checkpoints infeasible.
>
> That said, we do not want the work to be a “black box.” In the revised version, we will substantially strengthen method-level reproducibility, so that independent groups can reimplement AFM on their own industrial (or public) data. We have added a new Appendix A.4 with the following content:
>
> * Pseudocode for the full pipeline: per-sensor tokenization, event-channel construction, backbone training loop, uncertainty calibration, and inference/alarm logic.
> * A detailed list of all architectural and training hyperparameters: transformer depth/width, vocabulary sizes, quantization bins and clip radii, normalization scheme, window length/stride, optimizer, learning rate schedule, batch size, number of steps, and early-stopping criteria.
> * A description of the preprocessing and data schema, including how continuous tags and events are selected, normalized, clipped, and aligned, and how train/validation/test splits are constructed in time.
> * For the calibration and deployment layer, we spell out the exact procedures used for interval construction, isotonic regression, and dual-gate alarm thresholds, so that these can be reproduced independently of our data.
>
> Our intent is that, while the exact industrial corpus and pretrained weights cannot be shared for IP reasons, all modeling and training details are sufficiently specified that another group with access to multivariate industrial time series can reproduce our setup and validate the main claims on their own data.

---

> > ### Comment · Reviewer_KEX6 · 2025-11-21
> >
> > 1. For the first question, I do not mean I question your novelty. I mean, you need to improve your writing significantly for these points.
> > 2. For the second question, I read your Sec. 5.3 (NOT Sec. 5.2 thanks!), maybe you misunderstood my question. I mean, what about the performance of using some SOTA models, such as ChatGPT-5, Gemini? I know that these are not specific to this domain. But the performance of these models is really good, so they may achieve a good performance.
> > 3. For the third question, I can understand you, and I encourage you to do what you said in the rebuttal comments. But it's not good enough before you do that.

---

> > ### Comment · Reviewer_KEX6 · 2025-11-21
> >
> > Thanks for the author's response. The response answers some parts of my question, but not all of them. So I encourage the authors to further discuss. If the additional clarification is convincing, I will consider adjusting the score upward. Thanks!

---

> ### Author Response · Authors · 2025-11-20
>
> > What about the training cost in this paper?
>
> In regard to training costs, Appendix A.4.5 and A.4.6 may be of interest as we go into detail of the training resources and estimated carbon footprint. Please note that the exact training costs will vastly differ between setups due to varying factors (e.g., cloud provider pricing models, fixed-rate vs flex-rate costs, enterprise agreements for reduced rates, research credits, volume discounts). For this paper, our training expenditures measure around 1,000-2,000 USD, which include all experimentation and exclude deployment and miscellaneous costs.
>
> **We hope our revisions and responses address your concerns. If there are any additional questions or comments, we would be happy to address them.**

---

> > ### Comment · Reviewer_KEX6 · 2025-11-21
> >
> > Regarding the question “What about the training cost in this paper?”, thank you for your response. I believe your explanation is adequate for now and should not lead to any point deductions.

---

> ### Comment · Reviewer_KEX6 · 2025-11-24
>
> Thanks again for the authors' response! I encourage you to respond to the rest of my question as soon as possible. I will appreciate it and then determine the final score. Thanks!

---

> ### Author Response · Authors · 2025-11-24
>
> We apologize for the delay in response and thank you for your patience. Please see the responses to the additional inquiries below:
>
> > For the first question, I do not mean I question your novelty. I mean, you need to improve your writing significantly for these points.
>
> We understand and agree with your feedback. We have added Appendix A.5 (Novelty) to explain the novelty of AFM compared to SOTA models. We hope that in addition to the motivations provided in Sec. 1 (Introduction) and our results in Sec. 5 (Experiments), this will help readers better understand the contributions of our research.
>
> > For the second question, I read your Sec. 5.3 (NOT Sec. 5.2 thanks!), maybe you misunderstood my question. I mean, what about the performance of using some SOTA models, such as ChatGPT-5, Gemini? I know that these are not specific to this domain. But the performance of these models is really good, so they may achieve a good performance.
>
> 1\) Thank you pointing out the discrepancy. We have amended the baseline section reference in the responses.
>
> 2\) Although general-purpose large language models (LLMs) such as ChatGPT-5, Gemini, or Claude demonstrate impressive performance across natural language, reasoning, and multimodal tasks, their architectures and training regimes are not optimized for continuous, high-frequency industrial time-series forecasting. These models are primarily trained on internet-scale text and image corpora, which do not contain the physical, process-driven temporal patterns needed for accurate forecasting of pressures, flows, temperatures, or vibration signatures. As a result, they lack exposure to domain-specific statistical structures (e.g., equipment dynamics, thermodynamic relationships, or control-loop interactions) that are essential for producing stable and actionable predictions in oil and gas environments. Even when prompted with careful instructions and few-shot examples, LLMs tend to generate statistically plausible but physically inconsistent sequences (e.g., violating mass balance or equipment ramp-rate limits), which is unacceptable for operational forecasting or anomaly detection.
>
> A second fundamental limitation is scalability and latency. Multivariate industrial systems often involve hundreds of sensor channels sampled at high frequency across millions of timesteps. State-of-the-art LLMs are not designed to ingest such large structured matrices efficiently; their input token limits and quadratic attention cost make them prohibitively expensive when applied naively to long time-series windows. For example, a single inference using a 4k–32k context LLM cannot directly accommodate a 168-step x 100-sensor multivariate sequence without projection, flattening, or lossy compression. Even assuming such encoding were possible, the latency of querying a massive external LLM cannot meet the real-time requirements of industrial applications such as compressor surge detection, pump failure early warning, or plant-level alarm systems, where predictions must update every few seconds or minutes with strict reliability guarantees.
>
> Cost and operational risk also prohibit deployment of general-purpose LLMs in production. Running GPT-class or Gemini-class models continuously for streaming inference is far more expensive than hosting an in-house time-series model. Industrial telemetry streams rarely pause: a mid-size facility may produce >10 billion readings per year. Passing these streams to a third-party LLM service introduces significant recurring compute cost, data privacy concerns, and risk of service interruptions, all of which conflict with safety-critical reliability and strict data-governance requirements in oil and gas operations. Moreover, these LLMs behave as black-box models with limited control over internal state, calibration, and uncertainty quantification, making them unsuitable for regulated environments that require evidence-based alarm logic, reproducible behavior, or certification of AI models.
>
> In contrast, the AFM model is engineered specifically for industrial multivariate time-series: it encodes long temporal windows efficiently, handles high-dimensional sensor arrays, and includes quantization, calibration, and event-fusion mechanisms tuned for real equipment behavior. It runs at sub-millisecond latency on commodity GPUs or CPUs, can be deployed entirely on-premise, and provides controllable uncertainty thresholds essential for industrial alarm systems. Thus, while general-purpose LLMs excel at broad reasoning tasks, their training data, compute footprint, and inference characteristics make them ill-suited for continuous forecasting, anomaly detection, and decision support in field operations.

---

> > ### Author Response · Authors · 2025-11-24
> >
> > > For the third question, I can understand you, and I encourage you to do what you said in the rebuttal comments. But it's not good enough before you do that.
> >
> > We apologize for the misunderstanding. In the previous response for this question, we intended to convey that Appendix A.4 has already been added with all the mentioned reproducibility-related content (e.g., pseudocode, hyperparameters, calibration and deployment details, and more). If there are any additional questions or requests after review, we would be happy to address them.

---

### Meta-Review · Area_Chair_Q7Ce · 2026-01-02

**Summary:**

This work introduces Asset Foundation Model (AFM) for industrial asset performance management, built on a pre-trainable transformer backbone with per-sensor tokenization and fine-tuned on asset-specific historie. AFM is specifically designed to operate in industrial settings characterized by heterogeneous sensors and stringent real-time deployment requirements. The work also as a theoretical component with closed form bounds on quantization error and a Lipschitz stability result for discretization noise being proposed.

The paper received 3 reviews all of which leaned more on the negative side. The major points of contention from the reviewers were as follows:

* Majority of the reviewers questioned the overall novelty of the proposed foundation model with the major concern being the overall writeup emphasizing the application oreinted nature of the proposed foundation model rather than showing technical advancements. The authors agree to the same in the rebuttal and proposed to change the introduction to highlight the same. Overall, I think not every paper has to show a strong technical advancement if the underlying application is high impact. But the write up should not be confused between the two i.e. the paper being an application paper or a new method based work.

* The empirical evidence provided in the paper was also questioned with the reviewers not being satisfied by the baselines and number and quality of the experimental results.

**Reviewer Concerns:**

The authors provided a detailed rebuttal and tried to answer all the reviewer comments with concise yet concrete answers. New baselines were added (from the rebuttal it seems like Table 1 is new although I am not sure about this). Although these new experiments are appreciated, when I look at the results, I unfortunately do not see a need for a completely new set of foundation models for the underlying asset performance management. The previous statement is with respect to the paper rather than the field as a whole. The MAE and MSE reported are not very impressive wrt other reported baselines. So, I do not see a very strong empirical foundation that could be taken forward in future research. Thus, this is a major unresolved point from the reviews.

The novelty of the work is sufficiently addressed in the rebuttal.

**Reviewer Scores:**

Based on the rebuttal, a reviewer did raise a score and another one was in the favour of increasing the score. So based only on the text, this makes the paper a borderline case. Based on my assessment, I think the work requires a bit more re-structuring and more clarity to be considered for acceptance. As I have mentioned, the empirical results are not very strong when compared to the baselines and although I do get that this is a more generalized model and can handle more tasks, the writing leaves a lot to be desired. Another example are the figures which are not of great quality. I should not have to zoom 400% in order to see what is really going on.

I believe that the work tries to solve a real world problem and should be appreciated but it does not do a good job of convincing the readers of the same. I recommend rejection.

---

### Decision · Program_Chairs · 2026-01-26

Reject